# OpenNeRF: Open Set 3D Neural Scene Segmentation with Pixel-Wise Features and Rendered Novel Views

**Francis Engelmann**[1,2]**, Fabian Manhardt**[2]**, Michael Niemeyer**[2]**, Keisuke Tateno**[2]**,**
**Marc Pollefeys**[1,3]**, Federico Tombari**[2,4]
[1]ETH Zurich, [2]Google, [3]Microsoft, [4]TU Munich

## Abstract

Large visual-language models (VLMs), like CLIP, enable open-set image segmentation to segment arbitrary concepts from an image in a zero-shot manner. This goes beyond the traditional closed-set assumption, *i.e.*, where models can only segment classes from a pre-defined training set. More recently, first works on open-set segmentation in 3D scenes have appeared in the literature. These methods are heavily influenced by closed-set 3D convolutional approaches that process point clouds or polygon meshes. However, these 3D scene representations do not align well with the image-based nature of the visual-language models. Indeed, point cloud and 3D meshes typically have a lower resolution than images and the reconstructed 3D scene geometry might not project well to the underlying 2D image sequences used to compute pixel-aligned CLIP features. To address these challenges, we propose OpenNeRF which naturally operates on posed images and directly encodes the VLM features within the NeRF. This is similar in spirit to LERF, however our work shows that using pixel-wise VLM features (instead of global CLIP features) results in an overall less complex architecture without the need for additional DINO regularization. Our OpenNeRF further leverages NeRF's ability to render novel views and extract open-set VLM features from areas that are not well observed in the initial posed images. For 3D point cloud segmentation on the Replica dataset, OpenNeRF outperforms recent open-vocabulary methods such as LERF and OpenScene by at least +4.9 mIoU.

## 1 Introduction

3D semantic segmentation of a scene is the task of estimating, for each 3D point of a scene, the category that it belongs to. Being able to accurately estimate the scene semantics enables numerous applications, including AR/VR, robotic perception (Zhi et al., 2021; Zurbrügg et al., 2024) and autonomous driving (Kundu et al., 2022; Kreuzberg et al., 2022), since they all require having a fine-grained understanding of the environment. While the domain of 3D scene understanding has recently made a lot of progress (Schult et al., 2023; Choy et al., 2019; Yue et al., 2023; Qi et al., 2017; Takmaz et al., 2023b), these methods are exclusively trained in a fully supervised manner on (*closed-*)sets of semantic categories, rendering them impractical for many real world applications

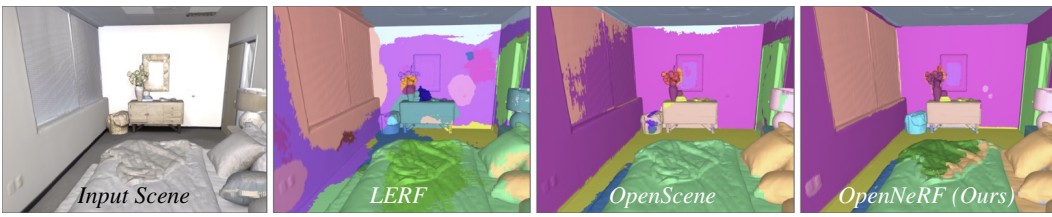

● Wall ● Ceiling ● Floor ● Blinds ● Window ● Lamp ● Door ● Pillow ● Cabinet ● Blanket ● Cushion ● Bed ● Picture ● IndoorPlant ● Shelf ● Vase ● Wall Plug ● Book ● Switch

**Figure 1:** Open-vocabulary 3D semantic segmentation on point clouds. Compared to LERF (Kerr et al., 2023), the segmentation masks of OpenNeRF are more accurate and better localized, while achieving more fine-grained classification than OpenScene (Peng et al., 2023). Zero-shot results on Replica (Straub et al., 2019).

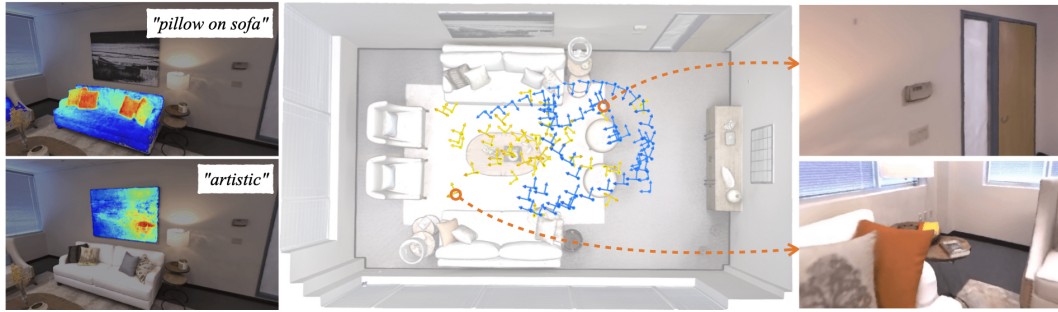

**Figure 2:** We propose OpenNeRF, an approach for open-set 3D scene understanding based on neural radiance fields. Arbitrary concepts can be queried from our representation *(left)*. As the original camera trajectory *(blue, middle)* might not capture all interesting scene details, we use NeRFs ability to render novel views *(right)* and propose a mechanism to obtain relevant novel camera poses *(yellow, middle)* that focus on scene details from which we can extract additional open-scene features improving the overall open-set scene representation.

as the models lack the flexibility to continuously adapt to novel concepts (Delitzas et al., 2024) or semantic classes (Dai et al., 2017a; Armeni et al., 2017; Ramakrishnan et al., 2021; Baruch et al., 2021; Straub et al., 2019). Therefore, in this work we aim at tackling the problem of *open-set* 3D scene segmentation. The main idea of *open*-set scene segmentation is that arbitrary concepts can be segmented, independent of any pre-defined closed set of classes. Given an *arbitrary* query – for example, a textual description or an image of an object – the goal is to segment those parts in the 3D scene that are described by the query. Such general unconstrained functionality can be crucial for helping robots interact with previously unseen environments, or applications on AR/VR devices in complex indoor scenes, especially when training labels are scarce or unavailable (Ahn et al., 2022).

Vision-language models (VLMs), such as CLIP (Wang et al., 2022) or ALIGN (Jia et al., 2021), have shown impressive performance on open-set image classification. Trained on internet-scale image-caption pairs, they learn a joint embedding mapping text and image inputs to the same (or different) embedding vector depending on whether they describe similar (or different) concepts. More recently, these powerful concepts were applied to dense pixel-level tasks, enabling open-set 2D image segmentation (Ghiasi et al., 2022; Li et al., 2022a; Rao et al., 2022). Using VLMs in combination with 3D point clouds is to date less explored. Similar to Li et al. (2022a), Rozenberszki et al. (2022) train a 3D convolutional network that predicts per-point CLIP features in a fully-supervised manner using the CLIP-text encodings of the annotated class names. Such fine-tuning on densely annotated datasets works well only on the classes labeled in the training set but does not generalize well to novel unseen classes which limits open-set scene understanding. OpenScene (Peng et al., 2023) recently proposed the first exciting results on open-vocabulary 3D scene understanding. In line with Rozenberszki et al. (2022), given a reconstructed 3D point cloud as input, a 3D convolutional network predicts CLIP features for each point. However, unlike Rozenberszki et al. (2022), the model is supervised with projected CLIP-image features from posed 2D images, preserving the generalization ability of the pixel-aligned visual-language image features. Similarly, LERF (Kerr et al., 2023) embeds CLIP features at multiple scales into neural radiance fields. As in Kobayashi et al. (2022), LERF adds a branch to predict CLIP features for a given 3D location. Subsequently, this enables rendering CLIP images, which is used to perform open-set semantic segmentation by means of computing the similarity with the CLIP feature from encoded input text queries.

While OpenScene (Peng et al., 2023), OpenMask3D (Takmaz et al., 2023a), and LERF (Kerr et al., 2023) demonstrate impressive capability of segmenting any given concept, they still suffer from several limitations. Exemplary, OpenScene fully operates on a given 3D scene reconstruction in the form of a polygon mesh (or a point cloud sampled from the mesh surface) that usually stems from a recorded RGB-D sequence. This approach is naturally limited by the mesh resolution, thus constraining the representation of smaller objects. On the other hand, LERF does not exhibit this limitation. However, as it relies on the CLIP-image encoder, the resulting global VLM features are not well localized in 3D space which leads to segmentation masks that a fairly inaccurate as CLIP can only be computed on full images or crops. To compensate for this issue, LERF employs multi-resolution patches for CLIP computation and further regularizes the rendered CLIP embeddings using DINO features. While this can mitigate inaccurate masks to some extent, it increases overall model complexity, yet achieving inferior open-set 3D segmentation masks (see Fig. 1).

In this work, we address the aforementioned limitations and propose OpenNeRF, a novel neural radiance field (NeRF) (Mildenhall et al., 2020) based approach for open-set 3D scene segmentation. As neural representation, NeRFs have inherently unlimited resolution and, more importantly, they also provide an intuitive mechanism for rendering novel views from arbitrary camera positions. Thus, we leverage this ability to extract additional visual-language features from novel views leading to improved segmentation performance. One key challenge is to determine the relevant parts of the scene requiring further attention. We identify the disagreement from multiple views as a powerful signal and propose a probabilistic approach to generate novel view points. Second, we further propose the direct distillation of pixel-aligned CLIP features from OpenSeg (Ghiasi et al., 2022) into our neural radiance field. Compared to LERF, this not only increases the segmentation quality via detailed and pixel-aligned CLIP features, it also significantly simplifies the underlying architecture as it eradicates the need for multi-resolution patches and additional DINO-based regularization terms.

Our experiments show that NeRF-based representations, such as LERF and ourOpenNeRF, are better suited for detecting small long-tail objects compared to mesh based representations used *e.g.*, in OpenScene. Further, we find that by incorporating pixel-aligned CLIP features from novel views, our system improves over prior works despite exhibiting a simpler design. We identify the Replica dataset as a promising candidate to evaluate open-set 3D semantic segmentation since, unlike ScanNet (Dai et al., 2017a) or Matterport (Ramakrishnan et al., 2021), it comes with very accurate mesh reconstruction, per-point semantic labels as well as a long-tail class distribution (see Fig. 4).

In summary, the contributions of this work are as follows:

- We propose OpenNeRF, a novel approach for open-set 3D semantic scene understanding based on distillation of pixel-aligned CLIP features into Neural Radiance Fields (NeRF).
- We propose a mechanism utilizing NeRF's view synthesis capabilities for extracting additional visual-language features, leading to improved segmentation performance.
- We present the first evaluation protocol for the task of open-set 3D semantic segmentation, comparing explicit-based (mesh / point cloud) and implicit-based (NeRF) methods.
- OpenNeRF significantly outperforms the current state-of-the-art for open-vocabulary 3D segmentation with an $+4.5$ mIoU gain on the Replica dataset.

## 2 RELATED WORK

**2D Visual-Language Features.** CLIP (Wang et al., 2022) is a large-scale visual-language model trained on Internet-scale image-caption pairs. It consists of an image-encoder and a text-encoder that map the respective inputs into a shared embedding space. Both encoders are trained in a contrastive manner, such that they map images and captions to the same location in the embedding space if the caption describes the image, and different locations in the opposite case. While the CLIP image-encoder yields a single global feature vector per image, LSeg (Li et al., 2022a) extends this idea and predicts pixel-level features which enables dense image segmentation tasks. Pixel-aligned CLIP features are obtained via fine-tuning on a fully-annotated 2D semantic segmentation dataset. This works well for the semantic classes present in the fully-annotated dataset, however, the pixel-aligned features generalize less well to novel concepts outside of the training classes. OpenSeg (Ghiasi et al., 2022) further improves on these aspects and proposes a class-agnostic fine-tuning to obtain pixel-aligned features. OVSeg (Liang et al., 2023) is the latest development that proposes to fine-tune a CLIP-like model on cropped objects. In this work, we use the pixel-wise features from OpenSeg (Ghiasi et al., 2022) which enables a direct and fair comparison with the publicly available models of OpenScene (Peng et al., 2023). Further, as also noted by (Peng et al., 2023), OpenSeg improves over LSeg (Li et al., 2022a) on long-tail classes unseen during the training of LSeg.

**Neural Radiance Fields.** Since the introduction of Neural Radiance Fields (NeRFs) (Mildenhall et al., 2020) for view synthesis, they have been adopted as scene representation for various tasks ranging from 3D reconstruction (Oechsle et al., 2021; Wang et al., 2021; Yu et al., 2022; Yariv et al., 2021) to semantic segmentation (Zhi et al., 2021) due their simplicity and state-of-the-art performance. Next to impressive view synthesis results, NeRFs also offer a flexible way of fusing 2D-based information in 3D. A series of works have explored this property in the context of 3D semantic scene understanding. In PanopticLifting (Siddiqui et al., 2023), predicted 2D seman-

tic maps are utilized to obtain a 3D semantic and instance-segmented representation of the scene. In (Kobayashi et al., 2022), 2D semantic features are incorporated and fused during the NeRF optimization which are shown to allow for localized edits for input text prompts. Neural Feature Fusion Fields (Tschernezki et al., 2022) investigates the NeRF-based fusion of 2D semantic features in the context of 3D distillation and shows superior performance to 2D distillation. Finally, in LERF (Kerr et al., 2023), NeRF-based 3D CLIP-feature fields are optimized via multi-scale 2D supervision to obtain scene representations that allow for rendering response maps for long-tail open-vocabulary queries. While all of the above achieve impressive 3D fusion results, we propose to investigate NeRF-based feature fusing in the context of open-set 3D scene segmentation. This does not only require to solve additional challenges, such as detecting relevant parts of the scene, but also enables more rigorous evaluation and comparison to other fusion approaches.

**3D Open-Set Scene Understanding** While most approaches for 3D scene understanding utilize 3D supervision (Han et al., 2020; Hu et al., 2021; Li et al., 2022b), a recent line of works investigates how 2D-based information can be lifted to 3D. In Semantic Abstraction (Ha & Song, 2022), 2D-based CLIP features are projected to 3D space via relevancy maps, which are extracted from an input RGB-D stream. While achieving promising results, their 3D reasoning is coarse and thus limited. Similarly, in ConceptFusion (Jatavallabhula et al., 2023) multi-modal 3D semantic representations are inferred from an input RGB-D stream by leveraging 2D foundation models. They use point clouds as 3D representation. ScanNet200 (Rozenberszki et al., 2022) uses CLIP to investigate and develop a novel 200-class 3D semantic segmentation benchmark. Most similar to our approach is OpenScene (Peng et al., 2023), which is the only existing method for open-set 3D semantic segmentation of point clouds. Our method is also similar to LERF (Kerr et al., 2023) in terms of NeRF representation but does not demonstrate results on 3D semantic segmentation. Further, our NeRF-based scene representation offers high-quality novel view synthesis which we leverage to render novel views of "interesting" scene parts towards improved segmentation performance.

## 3 METHOD

Given a set of posed RGB input images and corresponding pixel-aligned open-set image features, we want to obtain a continuous volumetric 3D scene representation that can be queried for arbitrary concepts. Our approach OpenNeRF enables open-set 3D scene understanding to address a wide variety of different tasks, such as material and property understanding as well object localization. By means of leveraging the normalized cosine similarity between the encoded queries and the rendered open-set features, we can explore multiple concepts (see Fig. 6). Alternatively, our approach can by seen as a method for unsupervised zero-shot 3D semantic segmentation (Fig. 5) where we assign to each point the semantic class with the highest similarity score.

### 3.1 OPEN-SET RADIANCE FIELDS

A radiance field is a continuous mapping that predicts a volume density $\sigma \in [0, \infty]$ and a RGB color $\mathbf{c} \in [0, 1]^3$ for a given input 3D point $\mathbf{x} \in \mathbb{R}^3$ and viewing direction $\mathbf{d} \in \mathbb{S}^2$. Mildenhall et. al. (Mildenhall et al., 2020) propose to parameterize this function with a neural network (NeRF) using a multi-layer perceptron (MLP) where the weights of this MLP are optimized to fit a set of input images of a scene. To enable higher-frequency modeling, the input 3D point as well as the viewing direction are first passed to a predefined positional encoding (Mildenhall et al., 2020; Tancik et al., 2020) before feeding them into the network. Building on this representation, we additionally assign an open-set feature $\mathbf{o} \in \mathbb{R}^D$ to each 3D point:

$$f_\theta(\mathbf{x}, \mathbf{d}) \mapsto (\sigma, \mathbf{c}, \mathbf{o}) \tag{1}$$

where $\theta$ indicates the trainable network weights. Our representation is based on Mip-NeRF (Barron et al., 2022) for appearance and density, while an additional MLP head models the open-set field. Essentially, the rendering of color, density and the open-set features is conducted following volumetric rendering via integrating over sampled point positions along a ray $\mathbf{r}$ (Levoy, 1990). We supervise the open-set head with OpenSeg feature maps which provide localized per-pixel CLIP features. As a result, we do not require rendering CLIP features at multiple scales for training or multiple renderings for each scale during inference as in LERF, leading to a simpler and more efficient representation.

## 3.2 TRAINING OBJECTIVES

**Appearance Loss.** Following standard procedure, we optimize the appearance using the Euclidean distance between the rendered color $\hat{\mathbf{c}}_r$ and the ground truth color $\mathbf{c}_r$ over a set of sampled rays $\mathcal{R}$:

$$\mathcal{L}_{\text{RGB}} = \frac{1}{|\mathcal{R}|} \sum_{r \in \mathcal{R}} ||\mathbf{c}_r - \hat{\mathbf{c}}_r||^2. \tag{2}$$

**Depth Loss.** While our method is able to be trained from RGB data alone, we can also leverage depth data if available. To this end, we supervise the density for those pixels where observed depth information is available (see Table 2 for an analysis). We compute the mean depth from the densities for each ray $r$ and supervise them using the smooth $L_1$ (Huber) loss:

$$\mathcal{L}_{\text{depth}} = \frac{1}{|\mathcal{R}|} \sum_{r \in \mathcal{R}} ||d_r - \hat{d}_r||. \tag{3}$$

**Open-Set Loss.** The open-set features are supervised via pre-computed 2D open-set feature maps from OpenSeg. Similar to Peng et al. (2023), we maximize the cosine similarity using

$$\mathcal{L}_{\text{open}} = \frac{1}{|\mathcal{R}|} \sum_{r \in \mathcal{R}} -\frac{\mathbf{o}_r}{||\mathbf{o}_r||_2} \cdot \frac{\hat{\mathbf{o}}_r}{||\hat{\mathbf{o}}_r||_2}. \tag{4}$$

Since the 2D open-set feature maps are generally not multi-view consistent, we do not backpropagate from the open-set feature branch to the density branch (Siddiqui et al., 2023; Kobayashi et al., 2022). Additionally, the 2D open-set maps from OpenSeg (Ghiasi et al., 2022) contain many artifacts near the image border. Therefore, we do not sample rays within a margin of 10 pixels from the image border during training. In summary, the total loss is defined as $\mathcal{L} = \mathcal{L}_{\text{RGB}} + \lambda_{\text{open}} \cdot \mathcal{L}_{\text{open}} + \lambda_{\text{depth}} \cdot \mathcal{L}_{\text{depth}}$.

## 3.3 RENDERING NOVEL VIEWS

A key advantage of NeRF-based representations is their ability to render photo-realistic novel views. These rendered novel views can naturally be used to extract 2D open-set features. We would like to use this ability to obtain improved open-set features for those parts of the scene where we have low confidence in the existing open-set features. A key challenge, however, is to first identify the parts of the scene that exhibit low confidence features and would thus benefit from rendering novel views.

**Confidence Estimation.** We identify the uncertainty $u_i \in \mathbb{R}$ over multiple projected open-set features as a surprisingly strong signal. Specifically, starting with the original color images, we compute open-set feature maps with OpenSeg. We then project each feature map onto a coarse 3D scene point cloud obtained using an of-the-shelf 3D reconstruction method (Dai et al., 2017b) and compute for each point $i$ the mean $\mu_i \in \mathbb{R}^D$ and covariance $\Sigma_i \in \mathbb{R}^{D \times D}$ over the per-point projected features. As per-point uncertainty measure $u_i \in \mathbb{R}$ we compute the *generalized* variance (Wilks, 1932) defined as $u_i = \det(\Sigma_i)$. Those parts of the scene that exhibit a large uncertainty intuitively correspond to areas where the open-set features from multiple viewpoints disagree on. They hence deserve further investigation by means of re-rendering from more suitable viewpoints. In practice, calculating the variance over a large number of high-dimensional features can be computational challenging. For a memory efficient and numerically stable implementation, we rely on Welford's online algorithm (Donald et al., 1999) to compute the variance in a single pass.

To confirm that idea, we compute the correlation between the per-point uncertainty $u_i$ and the per-point error $\epsilon_i$. We define the per-point error $\epsilon_i$ as the Euclidean distance between the ground truth open-set vector $\mathbf{o}_i^{\text{gt}}$ (*i.e.*, the CLIP text-encoding of the annotated class name) and the per-point mean open-set vector $\mu_i$. Indeed, measured over all Replica scenes, we observe a strong positive correlation ($\bar{r} = 0.653$) between $u_i$ and $\epsilon_i$. See Fig. 3 for an illustration.

**Novel Camera View Selection.** To generate novel camera poses we compute the `lookat` matrix based on a target $\mathbf{t} \in \mathbb{R}^3$ and camera position $\mathbf{p} \in \mathbb{R}^3$. As targets $\mathbf{t}$ we select points with high uncertainty (shown in red, Fig.3) based on their uncertainty $u_i$ if $x < u_i$ with $x \sim \texttt{Uniform}[0, 1]$. The camera position $\mathbf{p}$ is placed at a fixed small offset from the target position inside the scene with added small random noise. Importantly, we sample the density of the NeRF at the selected camera position to make sure the target is visible and does not collide with the scene geometry.

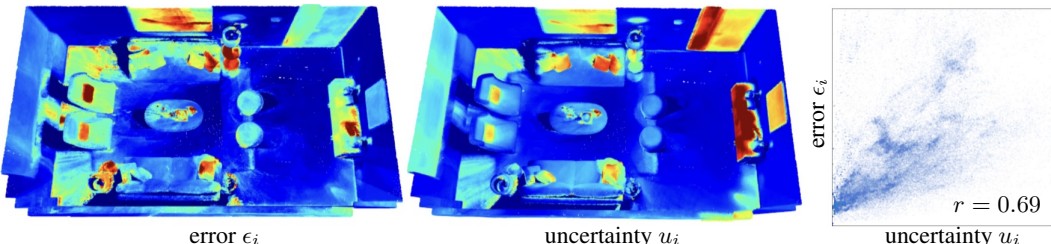

error $\epsilon_i$         uncertainty $u_i$         uncertainty $u_i$

**Figure 3: Confidence Estimation.** The error $e_i$ *(left)* correlates well with the estimated uncertainty $u_i$ *(center)*. Our mechanism for selecting novel view points is based on the estimated uncertainty. The plot *(right)* shows the error-uncertainty correlation $r$ for `room0` of the Replica (Straub et al., 2019) dataset.

## 3.4 Implementation and Training Details

Our model is implement in Jax based on Mip-NeRF (Barron et al., 2022). We train each NeRF scene representation for 3000 iterations and the pixel-aligned opens-set features are computed using OpenSeg (Ghiasi et al., 2022) resulting in $640 \times 360 \times D$ dimensional feature maps with $D = 768$. For memory efficiency reasons, we convert them to float16 values. For querying, we use the pre-trained CLIP text-encoder based on the ViT-144@336 model (Wang et al., 2022).

## 4 Experiments

**Datasets.** We evaluate our approach on the Replica (Straub et al., 2019) dataset and show qualitative results on scenes captured with the iPhone *3D Scanner App*. Replica consists of high quality 3D reconstructions of a variety of real-world indoor spaces with photo-realistic textures. Unlike other popular 3D semantic segmentation datasets, such as S3DIS (Armeni et al., 2017), Scannet (Dai et al., 2017a) or Matterport (Ramakrishnan et al., 2021), Replica is particularly well suited to evaluate open-set 3D scene understanding as it contains both a long-tail class distribution and carefully-annotated ground-truth semantic labels, including very small objects such as switches and wall-plugs. All experiments are evaluated on the commonly-used 8 scenes {`office0-4, room0-2`} (Zhu et al., 2022; Peng et al., 2023; Zhi et al., 2021), using the camera poses and RGB-D frame sequences from Nice-SLAM (Zhu et al., 2022). Each RGB-D video sequence consists of 2000 frames scaled to $640 \times 360$ pixels. For a fair comparison to OpenScene (Peng et al., 2023), we use only every 10-th frame for training resulting in 200 posed RGB-D frames per scene.

**Labels and Metrics.** Overall, the 3D scene reconstructions are annotated with 51 different semantic class categories. To enable a more detailed analysis of the experiments, we further split the original categories into three equally sized subsets (*head, common, tail*) based on the number of annotated points where each subset contains 17 classes (see Fig. 4). Note that the ground-truth semantic labels, however, are only used for evaluation and not for optimizing the representations.

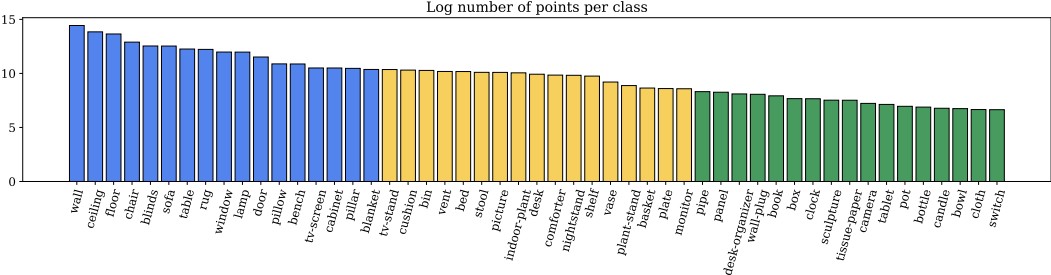

**Figure 4: Class Frequency Distribution of the Replica Dataset (Straub et al., 2019).** We show the number of point annotations for each category. The colors indicate the separation in *head* (blue), *common* (yellow) and *tail* (green) classes from left to right in decreasing order. Note that the plot is shown at log-scale.

|  | *All* | | *Head* | | *Common* | | *Tail* | |
|---|---|---|---|---|---|---|---|---|
|  | mIoU | mAcc | mIoU | mAcc | mIoU | mAcc | mIoU | mAcc |
| LERF (Kerr et al., 2023) | 10.5 | 25.8 | 19.2 | 28.1 | 10.1 | 31.2 | 2.3 | 17.3 |
| OpenScene (Peng et al., 2023) *(Distilled)* | 14.8 | 23.0 | 30.2 | 41.1 | 12.8 | 21.3 | 1.4 | 6.7 |
| OpenScene (Peng et al., 2023) *(Ensemble)* | 15.9 | 24.6 | 31.7 | 44.8 | 14.5 | 22.6 | 1.5 | 6.3 |
| OpenNeRF (Ours) | **20.4** | **31.7** | **35.4** | **46.2** | **20.1** | **31.3** | **5.8** | **17.6** |

**Table 1: 3D Semantic Segmentation Scores on Replica (Straub et al., 2019).** All results are obtained from image resolution of $640 \times 360$ pixels, using the OpenSeg image encoder and the CLIP text encoder based on ViT-144@336 and averaged over three runs. The reported LERF results are obtained using their original implementation adapted for Replica following the same experimental setup as ours. OpenScene scores are obtained with their provided pre-trained models. Note that the OpenScene models additionally profit from pre-training on the large-scale Matterport (Ramakrishnan et al., 2021) dataset.

For evaluation purposes, we compute the 3D semantic segmentation performance on the provided 3D scene point clouds. In particular, we follow (Peng et al., 2023) and measure the accuracy of the predicted semantic labels using the mean intersection over union (mIoU) and mean accuracy (mAcc) over all ground truth semantic classes and the *head*, *common*, *tail* subsets.

## 4.1 METHODS IN COMPARISON.

We compare our approach to the recently proposed OpenScene (Peng et al., 2023) and LERF (Kerr et al., 2023). OpenScene is currently the only method reporting open-world 3D semantic segmentation scores. For LERF, we obtain 3D semantic segmentation masks by rendering for each frame all relevancy maps over all evaluated semantic classes, then project these on the Replica point clouds and assign the semantic class with the highest summed relevancy score. The OpenScene model is a sparse 3D convolutional network that consumes a 3D point cloud and predicts for each point an open-set feature. The model is trained on large-scale 3D point cloud datasets and supervised with multi-view fused CLIP features. We compare with the two variations of their trained model, the 3D distilled model (Distilled), which directly predicts per-point features, and the improved 2D-3D ensemble model (Ensemble), which additionally combines the predicted per-point features with fused pixel features. For a fair comparison, we follow their experimental setup, using the same pixel-aligned visual-language feature extractor OpenSeg (Ghiasi et al., 2022). We use the public code and the Matterport (Ramakrishnan et al., 2021) pre-trained model as suggested in the official repository[1]. Note that OpenScene profits from additional training datasets not used by LERF or our OpenNeRF.

## 4.2 RESULTS ON 3D SEMANTIC SEGMENTATION

We show 3D semantic segmentation scores for all three subsets in Table 1. The 3D scene segmentations are obtained by querying the 3D scene representations for each one of the annotated ground truth semantic classes and assigning the class with the highest similarity score to each 3D point. Querying is performed via correlation of the 3D point cloud features with the embedding from large language models, specifically the CLIP-text encoder (ViT-L14@336). For OpenScene, we observe a similar trend as reported in (Peng et al., 2023), where the *Ensemble* model improves over the *Distilled* model (+1.1 mIoU).

Our results clearly improve over all baseline methods. We outperform LERF significantly while possessing a simpler overall design. Compared to OpenScene, we achieve +4.5 mIoU over all classes, +3.7 (head), +5.6 (common) and +4.3 (tail) for each subset. While achieving improved results on all classes, the smallest improvement can be found for the head classes. We attribute this to the fact that OpenScene can benefit from the pre-training on large-scale 3D datasets in this setting, enabling OpenScene to learn geometric priors of more popular classes (wall, ceiling, floor, chair). However, it is interesting to note that our approach is able to detect semantic categories that are not recognized at all by OpenScene (wall-plug, clock, tissue-paper, tablet, cloth). This is an important aspect, as these are often exactly the classes that are most relevant for an autonomous agent to interact with. Nevertheless, we also observe that numerous long-tail classes are not detected at all by neither method. This highlights that open-scene segmentation is a difficult task especially for long-tail classes, and shows that Replica is a challenging dataset for benchmarking.

---

[1] https://github.com/pengsongyou/openscene

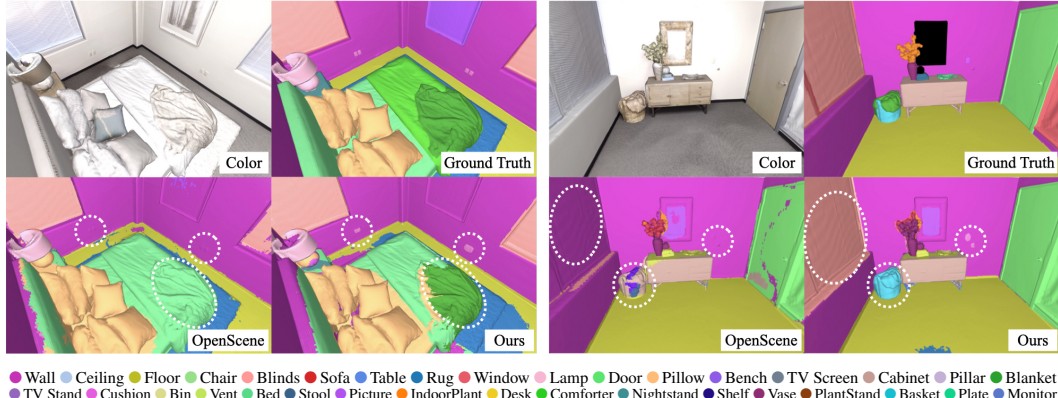

**Figure 5: Qualitative 3D Segmentation Results and Comparison with OpenScene (Peng et al., 2023).** The white dashed circles indicate the most noticeable differences between both approaches. Color and ground truth are shown for reference only. Overall, our approach produces less noisy segmentation masks.

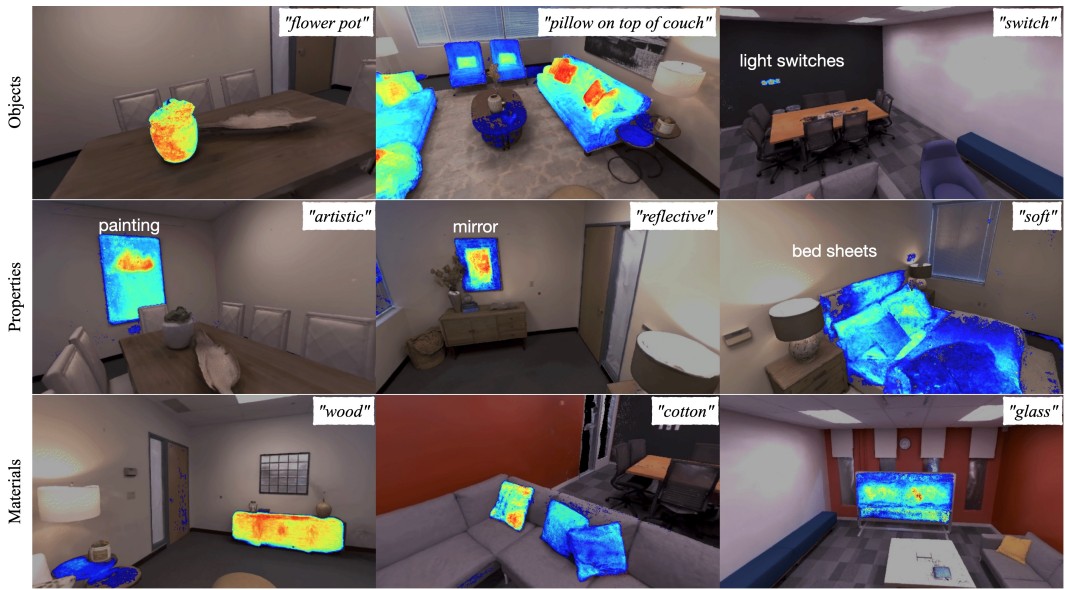

**Figure 6: Open-Set Scene Exploration.** We visualize the normalized cosine similarity between the rendered open-set scene features and the encoded text queries shown at the top-right of each example. Examples cover a broad range of concepts. Going beyond specific objects (*top*), we show scene properties (*middle*) and various materials (*bottom*). **Red** is the highest relevancy, **green** is middle, **blue** the lowest. Uncolored means the similarity values are under $0.5$.

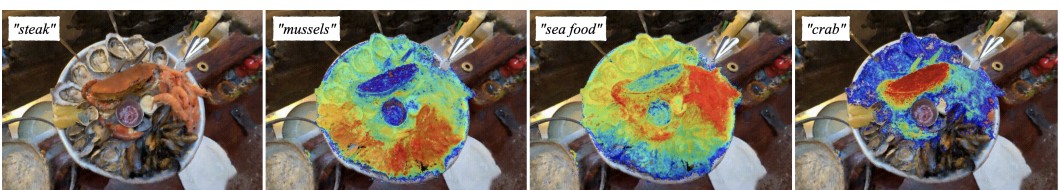

**Figure 7: In-the-wild Qualitative Results.** We capture scenes using the iPhone *3D Scanner App*, which provides camera poses from on-board SLAM and query the reconstructed scenes with various search terms.

### 4.3 Analysis Experiments

**Sampling or rendering?** Unlike OpenScene, which directly predicts per-point features for each point of a given 3D point cloud, our NeRF representation is more versatile. We can either directly *sample* ① the open-set features at a specified 3D position from the NeRF representation, or we can first *render* and then *project* ② the open-set features onto the 3D point cloud for evaluation. When multiple open-set features are projected to the same 3D point we take

|  | mIoU | mAcc |
|---|---|---|
| LERF (Kerr et al., 2023) | 10.4 | 25.5 |
| OpenScene (Peng et al., 2023) | 15.1 | 24.6 |
| ① Sampled | 16.5 | 29.1 |
| ② Render & Project | 18.5 | 29.8 |
| ③ + depth supervision | 19.4 | 30.8 |
| ④ + rendered novel views | **20.4** | **31.7** |

**Table 2: Analysis Experiments.**

the average over all points. Note that in both cases the 3D point cloud is only required for evaluation and, in contrast to OpenScene, not necessary to obtain the NeRF-based scene representation, which only relies on posed RGB(-D) images. Table 2 shows that the projection approach ② improves over sampling ①, which can be a direct consequence of the volumetric rendering within NeRFs that accumulates multiple samples along each ray compared to a single sample at a given 3D point. Note that ① already improves over both OpenScene and LERF.

**Impact of Depth Supervision.** We further analyze the importance of depth as additional supervision signal. We implement an additional regression loss using the Huber (smooth-$L_1$) loss between the rendered average distance and the ground truth depth from each training pose. Table 2, ③ shows that depth supervision further improves the open-set feature field since the additional depth supervision has a direct impact on the volumetric reconstruction quality.

**Impact of Novel Views.** Next, we analyze the contribution of the rendered novel views from the generated view points using the approach described in Section 3.3. Table 2, ④ clearly demonstrates the increased performance from rendered novel views. To disseminate whether the improvement comes from the novel views or simply from additional views, we also compare with views generated along the same camera trajectory as the original views which yielded the same results as ③. We then placed additional random cameras into the scene volume to render novel views from random positions. This results in a drastic performance drop (15.4 mIoU) due to numerous frames that are either inside the scene geometry or showing no meaningful context leading to deteriorated open-set features. This demonstrates the benefit of our novel view synthesis approach presented in Sec. 3.3.

### 4.4 Qualitative Results for 3D Scene Segmentation and Open-Set Applications

In Fig. 1 and 5, we compare qualitative semantic segmentation results of LERF, OpenScene and our OpenNeRF. The white dashed circles highlight the different predictions of each method. In contrast to OpenScene, our approach is able to correctly segment the wall-plugs as well as the blanket on the bed (Fig. 5, *left*). Our approach also properly detects the basket, the blinds and produces less noisy results on the door (Fig. 5, *right*). The more interesting aspect of open-set scene representation is that they can be queried for arbitrary concepts. In Fig. 6, we show the response of open-set queries. For each example, we provide the text query as label. We observe that our method can be used to not only query for classes, but also for concepts like object properties or material types. Finally, in Fig. 7 we show results on newly recorded *"in-the-wild"* sequences demonstrating the general applicability of our approach to unseen scenarios.

### 5 Conclusion

We presented OpenNeRF, a NeRF-based scene representation for open set 3D semantic scene understanding. We demonstrate the potential of NeRFs in combination with pixel-aligned VLMs as a powerful scene representation compared to explicit mesh representations, specifically on the task of unsupervised 3D semantic segmentation. We further exploit the novel view synthesis capabilities of NeRF to compute additional views of the scene from which we can extract open-set features. This enables us to focus in greater detail on areas that remain underexplored. To that end, we proposed a mechanism to identify regions for which novel views should be generated. In experiments OpenNeRF outperforms the mesh-based OpenScene as well as the NeRF-based LERF.

**Acknowledgements.** This work was partially supported by an ETH Career Seed Award funded through the ETH Zurich Foundation and an ETH AI Center postdoctoral fellowship.

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
