# OpenReview forum: "OpenNeRF: Open Set 3D Neural Scene Segmentation with Pixel-Wise Features and Rendered Novel Views"
_ICLR.cc/2024/Conference — ICLR 2024 poster_

### Official Review · Reviewer_AABJ · 2023-10-24

**Soundness:** 3 good
**Presentation:** 3 good
**Contribution:** 2 fair
**Rating:** 6
**Confidence:** 3

**Summary:**

This paper proposes to address the task of 3D open set detection by harassing implicit reconstruction methods, and specifically NeRF.
If I am not mistaken, the method boils down to adding supervision/prediction-head of per-pixel CLIP features in addition to the RGB/color head, while keeping the per-voxel density $\sigma$ prediction used in NeRF.
The authors use depth supervision, when available, as done in previous works.

The devil lies, as always, in the details of the training regime, and the authors stress some details that are important to make the network converge to the presented results:
The authors use OpenSeg feature maps which are more localized and therefore do not supervise them in a multi-scale fashion.
Additionally, one should refrain from using such features near the image edge as they are less stable.
Finally, one of the main points in the training process, the authors claim that training on novel views (generated by a trained NeRF) makes as big difference. Specifically, novel views are selected to minimize the uncertainty of the OpenSeg features, measured as generalized variance on the in 3D. This help mitigate areas where open-set features "disagree" from multiple viewpoints.

The experimental section evaluates the method compared to the previous 3D open-set approaches, which are all explicit.
The authors divide the labels into 3 groups by frequency, which better emphasized the strength of the propose method on rare ones.

**Strengths:**

* The paper was rather easy to follow, for an experienced NeRF reader (which is reasonable these days :-) ).
* The base of the method is rather straight forward to understand (adding CLIP predictions)
* The authors indicate uncertainty estimation of the open-set head is  a strong signal to help the network converge, with help of NeRF novel views.
* The results show the advantage of implicit methods over explicit ones
* The ablation study helps to understand which part of the training regime made the difference.

**Weaknesses:**

Many of the details of the paper were presented in previous works (e.g. per-pixel feature supervision was presented in NeRF-SOS fo DINO). and a big part of the novelty here is in the details of applying them to 3D segmentation.
This alone does not prevent acceptance, IMHO.

**Questions:**

A question on the novel-view based training:
Can you please comment on the seemingly bootstrapped nature of this approach?
In other words - a NeRF model has to be trained in the first place to generate novel views, and these novel views are then trained to predict open-set features?

---

> ### Author Response · Authors · 2023-11-21
>
> We thank the reviewer for the constructive and overall positive feedback.
>
> We are pleased that you appreciate two fundamental aspects of our work: The importance of open-set uncertainty estimation as a strong signal for guiding NeRF optimization, and the demonstrated advantage of our implicit method compared to explicit representations as used in OpenScene [Peng et al. 2023]. Furthermore, we appreciate your recognition that the core idea is straightforward to integrate into existing NeRF representations and that the paper is presented in a clear manner.
>
> Next, we address the remaining questions:
>
> - The reviewer asks if we first optimize a NeRF model to generate novel views. This is correct: the novel views stem from our optimized NeRF model, which is then further optimized with the new views. However, this strategy does not further complicate the training procedure, nor does it add computational overhead beyond the novel-view rendering. This is because the novel-views (and the corresponding pixel-level CLIP features) can be seamlessly integrated into the same optimization by simply adding them to the existing set of training images. Thank you for bringing this to our attention, we will enhance and clarify this aspect in the paper.
>
> - It is correct that details of this work have been addressed in previous research, for example [Kerr at al. 2023, Fan et al. 2022, Kobayashi et al. 2022, Zhi et al. 2021] etc. Given this, we appreciate your acknowledgment that the devil lies in the details, and that our novelty includes the combination of these distributed efforts into a single method for open-set 3D segmentation. On top of that, we are the first to directly lift pixel-level CLIP features to 3D within the scope of neural radiance fields. Further, as stressed earlier, a major contribution is how to leverage these CLIP features within open set 3D segmentation. To this end, we presented a novel confidence driven inference strategy and established a rigorous evaluation setup for the semantic segmentation task, demonstrating that our simpler yet more intuitive and faster setup leads to superior results (e.g., when compared to the more complex baseline of LeRF which requires multi-scale feature computation and additional DINO regularization terms).

---

> > ### Comment · Reviewer_AABJ · 2023-11-22
> > **Thanks for the detailed rebuttal.**
> >
> > The authors have addressed my questions, I am willing to increase my score to a positive one.

---

> > > ### Author Response · Authors · 2023-11-23
> > >
> > > We truly thank the reviewer for the quick reply and engaged rebuttal. Further, we would be happy if the reviewer could consider raising the score to clear accept score in light of the additional experiments and results in the rebuttal, to make the decision clearer and the final decision easier. Thank you very much.

---

### Official Review · Reviewer_UKPx · 2023-10-31

**Soundness:** 3 good
**Presentation:** 4 excellent
**Contribution:** 2 fair
**Rating:** 5
**Confidence:** 4

**Summary:**

The paper presents a NeRF-based scene representation for open-set 3D semantic scene understanding. The method extends previous work by leveraging pixel-aligned features and a mechanism to identify regions that require generating novel views. These additional views of the scene allow the proposed approach to extract more open-set features and improve the overall understanding of the 3D semantic scene. The results have been tested on the Replica dataset and have outperformed a mesh-based baseline and a NeRF-based baseline.

**Strengths:**

* The paper is well-written and justified. All the technical details are well-elaborated.
* The concept of using novel view synthesis capabilities to extract additional visual-language features for better scene understanding is interesting and straightforward.
* The qualitative results appear satisfactory and the quantitative results outperform OpenScene and LERF.

**Weaknesses:**

My main concern is regarding the evaluation:
* While the Replica dataset may be challenging for long-tailed settings, it would be great to evaluate on larger-scaled datasets and outdoor benchmarks such as Matterport3D, ScanNet, and nuScenes.
* One of the main contributions of this work is the use of novel view synthesis to extract additional visual-language features. To ensure the quality of the learned features, it seems necessary to evaluate the quality of the rendered views and their effect on scene understanding.
* To better evaluate the proposed method, it would be better to compare it with 2D methods such as LSeg, OV-Seg, and ODISE, as well as other 3D methods including Feature Field Distillation, Semantic-NeRF, and Panoptic Lifting (disregard whether these methods are open or closed-set).


Some questions regarding the confidence estimation and novel camera view selection:
* The uncertainty map in Figure 3 indicates that the door and cabinet are the most uncertain points, while the generated novel pose in Figure 2 seems to focus primarily on the table. Given high uncertainty points, how are the camera views sampled?
* Although there is a positive correlation between per-point uncertainty and per-point error, the heat maps appear to emphasize all areas other than the wall and floor. It is also intriguing that the peaks of uncertainties don't correspond to object boundaries, but rather the entire object. It would be better to demonstrate more samples to showcase these findings.

**Questions:**

Please see the weaknesses part for my concern and confusion.

---

> ### Author Response · Authors · 2023-11-23
>
> We thank the reviewer for the constructive and insightful feedback. Below, we address the remaining questions:
>
> __Experiments on Larger-Scale Datasets__
> We provide additional results on the ARKitScenes [Baruch et al. NeurIPS’21] dataset in the attached supplementary material. ARKitScenes is a large-scale, real-world dataset providing posed RGB-D video sequences from an iPhone/iPad device along with 3D reconstructions. We focus on long-tail, in the wild objects that are small and typically hard to detect and segment. As explained in LERF, we found ScanNet to be challenging to optimize high-quality NeRFs due to limited views and poor poses. However, more recent datasets, such as ARKitScenes, do not exhibit these problems. In fact, as shown in Fig.7, the posed RGB-D recordings from an iPhone can directly be used to optimize our NeRF representation.
>
> __Confidence Estimation and Novel Camera View Selection__
> The reviewer comments on the uncertainty visualized in Fig.3 and the selected camera poses shown in Fig.2, and raises the question why most of the selected camera poses (yellow) are around the table in the center of the scene, instead of the door or cabinet which exhibit larger uncertainty. We agree with you that the illustrations is a bit misleading;
> instead of the camera pose, we should visualize the camera frustum to show more clearly what is visible from the selected camera. Indeed, the current illustration might give the impression that the novel camera poses around the table are also looking at the table. However, this is not the case: what the cameras are looking at is (i.e., the camera target) is selected based on the estimated uncertainty. The actual camera position is placed at a distance from the target (to make sure the object and its context is visible) while making sure that the camera itself is not colliding with the scene geometry (see Sec.3.3 for details).
>
> __Object Boundaries vs. Entire Objects__
> The reviewer is curious why entire objects instead of object boundaries are emphasized in the uncertainty visualization (Fig.3). Uncertainty at object boundaries is an effect that is typically observed when visualizing the entropy of semantic-class probability distributions from a trained model. This means that the model is confident about the object label, but less sure about the object mask boundaries. In such a setup, the model acts as a regularization over the training data.
>
> Instead, Fig.3 shows the variance over open-set features computed independently for each view. These features originate from OpenSeg, an approach that provides pixel-aligned CLIP features based on a segmentation of the input image. When computing the uncertainty, there is no regularization across views, such that these features can disagree significantly. This explains why entire objects are highlighted in the heatmap instead of only object boundaries.
>
> _We hope that you find our answers convincing, and that you might find additional insights in the discussions with the other reviewers._

---

### Official Review · Reviewer_Y2vi · 2023-11-01

**Soundness:** 3 good
**Presentation:** 4 excellent
**Contribution:** 3 good
**Rating:** 8
**Confidence:** 5

**Summary:**

This paper introduces OpenNeRF, an innovative approach for superior open-set 3D semantic scene understanding. OpenNeRF demonstrates its suitability for detecting small long-tail objects, surpassing mesh-based representations in performance. Additionally, OpenNeRF effectively represents interesting scene parts, leading to improved segmentation performance.

**Strengths:**

1. The proposed OpenNeRF approach significantly improves 3D semantic segmentation results compared to baseline methods such as LERF and OpenScene. It can also detect challenging long-tail classes ignored by other methods.
2. Confidence estimation and analysis of novel views bring new insights for improving the Open-set 3D scene understanding.
3. OpenNeRF can be queried for arbitrary concepts, including object properties (eg. reflective, soft) and material types (eg. wood, cotton), showcasing its versatility. It shows impressive results in the wild scene using a phone scanner.

**Weaknesses:**

1. The performance drop when rendering novel views from random positions suggests that the approach is sensitive to the quality and meaningful context of the additional views.

**Questions:**

1. Could you provide more details on the evaluation protocol used to compare explicit-based (mesh/point cloud) and implicit-based (NeRF) methods for open-set 3D semantic segmentation? What were the key findings of this evaluation?
2. Have you considered the scalability and computational efficiency of OpenNeRF? How does it perform in terms of runtime and memory requirements compared to other methods?

---

> ### Author Response · Authors · 2023-11-21
>
> We thank the reviewer for the constructive and positive feedback.
>
> In particular, we are pleased that you recognize the performance of our model for open-set 3D semantic segmentation compared to the baselines, and specifically for long-tail classes, including our novel-view rendering based on confidence estimation that brings new insights for improving open-set 3D scene understanding. We are also happy, that you are convinced by the showcased results on arbitrary concept queries, even on in-the-wild-scene recorded with a commodity phone lidar scanner.
>
> Next, we address the remaining questions and concerns:
>
> __Runtime and Memory__
> The reviewer is asking about computational requirements of our method compared to other models. We measure the runtime and memory requirements for our model and compare it to LERF. For a fair comparison, we report all measurements on the same hardware and settings (image resolution, feature backbone, etc.).
> The table below shows the results:
>
> | Method 	| Model Parameters	| Avg.Runtime per Frame  | mIoU |
> | --- | --- | --- | --- |
> | LERF 		| 234 million 		| 22.6			| 10.5 |
> | OpenNeRF	|   84 million 		| 3.4 		 	| 20.4 |
>
> We look at the number of model parameters and the average runtime to perform a query. Our model relies on approx. 3 times fewer parameters than LERF while being 6-7 times faster and performing 2 times better on segmentation. These significant differences are largely due to our less complex pipeline: our model does not require the additional head for DINO regularization, our CLIP branch is significantly smaller than LERFs which implements a multi-scale feature field. In terms of runtime, for each prompt, LERF needs to compute features at multiple scales to select the best one, and additionally performs negative-prompt  to suppress background noise. We also integrated negative prompts into our model, but did not find them to be helpful in terms of performance. We are grateful for your input and will add this table to the updated version of the paper.
>
> __Details and Observations during Evaluation__
> Next, the reviewer is curious about details of the evaluation protocol and observations made during the evaluation. Specifically, we follow the exact same evaluation protocol as OpenScene (Sec.4), and compute the 3D semantic segmentation performance on the provided 3D scene point clouds. For NeRF based methods (ours and LERF), we aggregated projected features on a per-point basis to obtain the final semantic class (as described in Sec. 4.1 and 4.2). The interesting observation here is that directly sampling the NeRF representations at 3D point positions does not perform as well as the rendering and aggregating strategy. This can be explained by the fact that NeRFs are optimized in the projected image space. We compared these two alternatives in Sec. 4.3 (“Sampling or rendering?”).
>
> __Baseline Experiment__
> Finally, the reviewer mentions the performance drop in one of the baseline experiments, when rendering novel views from *random* positions (Sec.4.3, “Impact of Novel Views”). In the above review, this aspect is listed as a weakness of our method. However, only to clarify, this is not our method, but merely a baseline experiment. In fact, this analysis shows the benefit of our proposed uncertainty-guided camera pose selection over  randomly selecting poses. We will make this clearer in the paper.
>
> We hope that we were able to answer all your questions and are open to further clarifications if needed.

---

> > ### Comment · Reviewer_Y2vi · 2023-11-22
> >
> > Thanks for the author's response. My concerns have been essentially addressed.

---

### Official Review · Reviewer_Nxv7 · 2023-11-02

**Soundness:** 3 good
**Presentation:** 3 good
**Contribution:** 2 fair
**Rating:** 5
**Confidence:** 4

**Summary:**

This work explores a similar paradigm to LeRF (Language-embedded radiance fields). Rather than distilling CLIP features to NeRF, rather it uses a different backbone i.e. OpenSeg to distill OpenSeg features to the 3D using NeRFs. The idea of distilling similar open-set features to 3D has been shown by OpenScene which required 3D meshes as input 3D representation, however, the paper studies it in the setting of Neural Radiance Fields which requires posed 2D images as input. Qualitative open vocabulary segmentation comparisons are shown on the Replica dataset coupled with quantitative comparisons with baselines like LeRF and OpenScene

**Strengths:**

The approach discussed very relevant problems i.e. distilling open-set 2D features to the 3D domain and moving away from traditional pipelines that work with a known number of categories i.e. approaches like Semantic-NeRF and Panoptic NeRF etc. The main benefits/strengths of the approach are as follows:

1. Qualitative comparison with LeRF and Open-Scene clearly shows better segmentation results despite the fact that LeRF was not designed for segmentation (which should be clearly pointed out)

2. Clearly better quantitative results on all, head, common, and tail sets

3. A good measure of uncertainty to improve the open-set feature field distillation into 3D.

**Weaknesses:**

1.  Though the uncertainty measure is sound, I wonder if this improves LeRF and OpenScene's performance as well. A similar measure was introduced in Semantic-NeRF and the authors didn't highlight the difference between their formulation and Semantic NeRF's formulation.

2. This looks like an incremental work that extends LeRF by using a different encoder backbone which is very straightforward to implement. Can the authors justify it with really good segmentation performance on long tail in-the-wild queries etc? I didn't see that comparison

3. Not many in-the-wild examples could be seen in the paper. Replica dataset is easier since we have perfect viewpoint annotations. I wonder if the performance stays steady or breaks for more in the wild examples where there are imperfect viewpoint annotations.

4. Does uncertainty measure really help? What if the initial NeRFs are not that good? Features tend to break for those viewpoints/areas. Do the authors have improved results in those cases?

**Questions:**

Please see all the questions in the weakness section. Overall I would have liked to see a thorough comparison with LeRF and what technical improvement the work bring (in addition to changing the pre-trained backbone), a comparison with Semantic-NeRF's uncertainty formulation, more in the wild examples and examples where the quality of NeRF is not that good and do the semantic features break or does uncertainty help in those areas

---

> ### Author Response · Authors · 2023-11-23
>
> We thank the reviewer for the constructive and detailed feedback. In particular, we appreciate the recognition of the improved 3D segmentation results of our method over the baselines LERF and OpenScene on all sets (head, common, tail) as well as the proposed uncertainty to measure to improve open-set 3D scene understanding.
>
> Below, we address the remaining questions and concerns:
>
> __Comparison with LERF__
> The reviewer asked for a detailed comparison to the LERF baseline as shown in the paper. Below we highlight the most important aspects:
>
> 1) First, OpenNeRF proposes an overall *simpler* model (with fewer components that are easier to tune) which results in notably faster and improved scene understanding performance (see Table 1 below). In particular, the key differences in the model are:
> - The usage of pixel-aligned open-set features (from OpenSeg) as opposed to per-image features (from CLIP). This results in better localized open-set features as evidenced by our improved segmentation masks.
> - A direct consequence is that we do not need complex multi-scale mechanisms for feature representation as in LERF. Therefore our CLIP head is significantly smaller, since it does not need to represent a multi-scale feature field. Importantly, during inference, LERF requires a rendering of each scale, which linearly slows down the querying process by the number of scales. This effect is further emphasized by the negative prompts proposed in LERF to suppress background noise, which did not help seem to bring improvements in our setup.
> - Finally, our model does not depend on additional regularization terms; LERF requires an additional head for DINO regularization to keep the features consistent. In our setting, DINO regularization did not make a significant difference due to the pixel-aligned features.
> Overall, our simplified design avoids the representation of multi-scale CLIP and DINO features, which results in an overall slimmer and faster model.
> | Method 	| Model Parameters	| Avg.Runtime per Frame  | mIoU |
> | --- | --- | --- | --- |
> | LERF 		| 234 million 		| 22.6			| 10.5 |
> | OpenNeRF	|   84 million 		| 3.4 		 	| 20.4 |
>
> _Table1: Our model relies on approx. 3 times fewer parameters than LERF while being 6-7 times faster and performing 2 times better on segmentation._
>
>
>
> 2) Second, beyond the actual NeRF model as scene representation, a key aspect of the work is the demonstrated use of rendered novel views, based on the proposed uncertainty estimation, as a promising approach to extract refined open-set features. In fact, the reviewer pointed out an additional interesting experiment, to analyze the performance of baselines models using the proposed novel-view rendering technique. We integrated it into the LERF pipeline as shown in Table 2 below. Interestingly, the relative performance boost due to the novel views appears even more significant for LERF.
>
> | Method | LERF| + NovelViews | OpenNeRF | + NovelViews |
> | --- | --- | --- | --- | --- |
> | mIoU | 10.4 | 12.1 | 19.4| 20.4 |
>
> _Table 2: Effect of adding uncertainty-based novel-views to LERF and OpenNeRF (metric is IoU)._
>
> Since this mechanism depends on the novel-view rendering capabilities of NeRF, it can be easily integrated into NeRF-based methods such as LERF. However, it is a bit harder for OpenScene which does not support rendering of novel views. Nevertheless, one could imagine an extension of OpenScene where the provided textured meshes are rendered from novel camera poses based on our uncertainty measurement. This would then require the regeneration of the image-based training data as well as training/fine-tuning the 3D sparse convolutional Minkowski networks to obtain the improved 3D open-set features.

---

> ### Author Response · Authors · 2023-11-23
>
> __Additional “In-the-wild” Examples__
> The reviewer would like to see additional results from real-world scenes recorded in-the-wild. Unlike Replica,  real-world recordings typically do not have perfect viewpoint annotations which can make training of NeRFs harder. In the __attached supplementary we show additional “in-the-wild” real-world results__ on the ARKitScenes dataset which consists of iPhone RGB-D recordings with imperfect poses in natural environments. Specifically, we tried to pick “long-tail” objects, such as “door handle”, “remote control” etc. that are hard to localize and segment.
>
> __Comparison with SemanticNERF’s Uncertainty Formulation__
> The reviewer asks for a comparison of our confidence estimate (Sec.3.3) and a related uncertainty formulation in SemanticNeRF. Thanks a lot for this question, this aspect is less highlighted in the paper but nevertheless important.
> In our work, we compute the _generalized variance_ over high-dimensional OpenSeg/CLIP features projected to a single 3D point. SemanticNeRF computes the _entropy_ of the per-semantic-class probability distribution for each 2D pixel.  Both metrics are used to quantify uncertainty, however what they measure is different:
> - The entropy in SemanticNeRF measures the confidence of the semantic class prediction. Ideally, there would only be one peak for a single class in the predicted distribution. However, if the semantic prediction is uncertain, the probabilities will be spread out over multiple classes, resulting in larger entropy. SemanticNeRF uses this measure to analyze the effect of noisy and sparse supervision (Fig.5 and Fig.6 in SemanticNeRF).
> - Since our OpenNeRF (and also LERF) operates in the open-vocabulary setting, we cannot assume to know all existing semantic classes in advance, i.e., we cannot compute the entropy over class-probabilities. However, we can measure the variance of the open-set features in the high-dimensional space. Ideally, for each point in 3D space the projected open-set features would be equal. The more they disagree, the more they are distributed around their mean. As a measure for uncertainty, we therefore compute the multi-dimensional variance, which is summarized in a single scalar by the generalized variance used in this work. We use this single scalar to select novel camera views.
>
> Unlike SemanticNeRF which computes the entropy only for analyses, in our work, the generalized variance guides the novel-view selection mechanism used to improve the open-set features.
>
>
> __Dependence on NeRF quality__
> The reviewer wonders if our mechanism based on novel-views still works if the rendering quality of the NeRF deteriorates. As correctly pointed out, our method strongly depends on the quality of the rendered novel views. In case of low-quality novel-views, the extracted open-set features will naturally not be helpful. This is a clear limitation of our method that we will further discuss in the paper. A similar observation is made in LERF, which equally depends on high-quality views to train a NeRF.
>
> _We hope that the additional experiments, results and explanations are helpful for your final decision._

---

### Meta-Review · Area_Chair_zHDo · 2023-12-11

**Metareview:**

The paper introduces a novel scene representation technique based on Neural Radiance Fields (NeRF) for open-set 3D semantic scene understanding. This approach builds upon previous work by incorporating pixel-aligned features and introducing a mechanism for identifying areas that necessitate the generation of new views. Notably, the method demonstrates superior performance in benchmarks characterized by a long-tail distribution of objects. The reviewers have recognized the relevance and timeliness of this research in the current landscape. While there are some concerns regarding the originality of the techniques employed, these do not significantly detract from the potential acceptance of the paper, as per the assessment of the area chair. The overall sentiment is that the method's advancements and results could provide a meaningful contribution to the field, despite the noted reservations about its novelty.

**Justification For Why Not Higher Score:**

There are concerns over the novelty of the technique.

**Justification For Why Not Lower Score:**

The work is among the earliest to extend LeRF for the open set setup.

---

### Decision · Program_Chairs · 2024-01-16

Accept (poster)